# Conductive Polymer Nanoparticles as Solid Contact in Ion-Selective Electrodes Sensitive to Potassium Ions

**DOI:** 10.3390/molecules28073242

**Published:** 2023-04-05

**Authors:** Hui Bao, Jin Ye, Xuyan Zhao, Yuan Zhang

**Affiliations:** 1College of Information Science and Engineering, Henan University of Technology, Zhengzhou 450001, China; 2College of Electrical Engineering, Henan University of Technology, Zhengzhou 450001, China

**Keywords:** ion-selective, conductive polymer, carbon nanotubes

## Abstract

A preparation method of nanocomposites based on poly (3-octylthiophene-2,5-diyl) (POT) and carbon black (CB) as the transducer of an all-solid potassium ion selective electrode is proposed. POT is used as the dispersant of CB, and the obtained nanocomposites have unique characteristics, including high conductivity, high capacitance and high stability. The potassium ion selective electrode based on POT and CB was characterized by cyclic voltammetry (CV), electrochemical impedance spectroscopy (EIS) and chronopotentiometry. The results showed that the detection limit of potassium ions was 10^−6.2^ M, and the slope was 57.6 ± 0.8 mV/façade. The water layer test and anti-interference test show that the electrode has high hydrophobicity, the static contact angle reaches 139.7° and is not easily affected by light, O_2_ and CO_2_.

## 1. Introduction

An all-solid ion selective electrode (SC-ISE) is a favorable substitute for a classical ion selective sensor with an internal solution. The introduction of a conductive polymer (CP) as an ion–electron transducer improves the stability of the electrode potential [1]. In such applications, a significant advantage of a conductive polymer is that it mixes the electronic and ionic conductivity and can realize reversible charge transfer at all interfaces of the sensor. These transducer materials provide attractive characteristics for sensors: stable potential readings, reproducible potentials between different sensors and the possibility of customized analytical performance [2,3,4,5]. Because conductive polymers can conduct ions and electrons when properly doped, they can be used as effective ion–electron sensors through oxidation–reduction reactions. Polypyrrole [6], polyaniline [7], poly (3-octylthiophene) (POT) [8] and poly (3,4-ethylenedioxythiophene) (PEDOT) [9] are the most studied conductive polymers in SC-ISE. They can be deposited by electropolymerization or drop casting from a polymer solution. Polyaniline, polypyrrole and PEDOT show high stability, conductivity and redox capacitance. However, they all have electrical activity in a wide range of potentials, which will lead to some side reactions and subsequent potential drift. Poly (3-octylthiophene) (POT) is a highly hydrophobic CP with high hydrophobicity and easy deposition (drop casting or electrodeposition) on the electrode surface, and the participation of POT in the side reaction is much less. However, POT also has disadvantages, that is, its redox capacity and conductivity are much lower [5,10].

In recent years, carbon nanocomposites, such as porous carbon, carbon microspheres, carbon nanotubes, graphene, etc., have been widely used in the solid contact layer of all-solid-state ion selective electrodes [11,12,13,14,15,16,17] due to their unique physical and chemical properties to improve their sensitivity, potential stability and detection limit.

Although it has many advantages, the double-layer capacitance between the electrode substrate and the ion-selective membrane of the all-solid-state ion-selective electrode is small [18], and the charge transfer resistance is large [19]. It is easy to cause the electrode to produce a water layer [20,21], which leads to poor stability and electrode potential drift. When the concentration of the detection solution is low, the linear response is not ideal [22,23]. In view of these shortcomings, this study combined the high hydrophobicity of conductive polymers and the high specific surface area of carbon materials to prepare a composite material, which can be used as an ion–electron converter in the ion-selective electrode.

Carbon black (CB) is a carbon material that forms small particles (fused together) organized in a chain structure. CB has a semi-graphite microstructure, which is made of stacked nano-graphite embedded in amorphous carbon. CB has high porosity. Without any additional treatment, it has been proven to show superhydrophobic behavior [24,25,26]. CB has excellent conductivity, cost efficiency, easy to obtain stability and uniform dispersion, and is worthy of being widely used in the development of nano-modified electrochemical sensors. The modification of carbon black on the electrode surface can be used to improve the performance of the sensor and induce catalytic and/or electrocatalytic effects. Compared with other carbon-containing materials such as graphene and carbon nanotubes, CB is extremely cheap and exhibits similar or better electrochemical properties. Compared with other carbonaceous dispersions, CB dispersions also have higher stability and good resistance to the interference of oxygen, carbon dioxide and light [27,28,29,30]. CB dispersion is stable for at least two weeks, while multi-walled carbon nanotubes and graphene dispersion are no longer uniform on the second day after production. In addition, the carbon black modified electrode is characterized by high storage stability and can be stored for at least three months at room temperature and in dry conditions.

In this work, a method based on poly (3-octylthiophene-2,5-diyl) (POT) and carbon black (CB) nanocomposites as a solid-state selective electrode ion electron transducer was proposed. On this basis, the all-solid potassium ion selective electrode was prepared. It is characterized by cyclic voltammetry, chronopotentiometry and an impedance spectrum, and the anti-interference test of the water layer is also carried out.

## 2. Results and Discussion

### 2.1. Morphological Characterization

It can be seen from Figure 1a that the solid contact layer of CB has uniform morphology, relatively uniform distribution, flaky structure and large specific surface area, which can effectively enhance the ion adsorption capacity and increase the electrode potential stability. Figure 1b shows that the poly (3-octylthiophene-2,5-diyl) (POT) in THF is used to disperse CB, thus producing a time-stable suspension. For the POT-CB solid contact layer, there is obvious pore structure inside, forming a diameter close to a 1 μM spherical structure, which increases the surface roughness.

The hydrophobicity of the solid contact layer membrane is tested by the contact angle (CA) measurement method. A total of 5 μL of deionized water is dropped on the electrode surface and the static contact angle formed is observed. Figure 1c shows the static contact angle of CB-ISM and POT-CB-ISM. The static contact angle of the ion-selective membrane reached 139.7° when POT was fused with CB as a dispersant. This is mainly due to the super-hydrophobicity of POT itself and the nature of easy deposition on the electrode surface. After being combined with CB, it forms a three-dimensional connected porous structure with increased porosity and a more solid structure.

### 2.2. Electrochemical Performance Characterization

It can be seen from Figure 2 that the capacitive current of GC/K^+^-ISE is very small and there is no obvious redox peak. The capacitive current of GC/CB/K^+^-ISE is higher than that of GC/K^+^-ISE. The high specific surface area of CB increases the adsorption capacity of the electrode. GC/POT-CB/K^+^-ISE can observe the maximum anode and cathode current, mainly because the POT-CB has a three-dimensional connected porous structure, which improves the conduction path of the electrode surface and so it can improve the electron transfer ability between the electrode and the material.

The electrochemically active surface area has a significant impact on the electrochemical behavior. We calculate the electrochemically active surface area of the electrode using the Randles–Sevcik equation (Equation (1)) [31]:(1)$$I_{\mathrm{pa}}=\left(2.69\times 10^{5}\right)A_{e}n^{3/2}D_{0}^{1/2}C_{0}v^{1/2},$$
where $I_{\mathrm{pa}}$ is the peak current, n is the number of transferred electrons, *D*_0_ is the ferricyanide diffusion coefficient (6.3 × 10^−6^ cm^2^ s ^−1^) [32], *C*_0_ is the concentration, *v* is the scanning speed and *A*e is the electrochemically active surface area. Ferricyanide involves one electron transfer, *n* = 1. Through calculation, the EASA value of GC/POT-CB/K^+^-ISE is 0.11 cm^2^, 1.57 times the EASA value of GC/CB/K^+^-ISE (0.07 cm^2^), and 2.2 times the EASA value of bare electrode (0.05 cm^2^). The higher EASA value indicates that GC/POT-CB/K^+^-ISE has a higher electrochemically active surface area, which may be due to the fact that the POT-CB structure can provide more electrochemically active sites. Therefore, it can enhance the current response and help improve the electrocatalytic activity.

### 2.3. Potentiometric and Chronopotentiometric Measurements

The potential response of GC/POT-CB/K^+^-ISE was measured in KCl solution with a concentration of 1.0 × 10^−8^–1.0 × 10^−3^ M. It can be seen from Figure 3a that during the whole measurement process, when the solution concentration changes, the electromotive force value tends to be stable within 10 s, indicating that the response time of the prepared electrode is less than 10 s, which is due to the good catalytic activity of POT-CB. Figure 3b shows that GC/POT-CB/K^+^-ISE has a linear response in the range of potassium ion concentration of 10^−6^–10^−1^ M, with a linear correlation of R^2^ = 0.998. The response slope is 57.6 ± 0.8 mV/facade, which conforms to the Nernst response characteristics, and the detection limit of the electrode for potassium ion activity is 10^−6.2^ M. Compared with other electrodes and detection methods (Table 1), the results show that the prepared electrode has a lower detection limit and a larger linear range and is a promising electrochemical sensor.

### 2.4. Reverse Chronopotentiometric Measurement

Reversed-current chronopotentiometry is used to evaluate the capacitance of the solid-state ion selective electrode and the short-term potential stability of the developed electrode. In this work, reverse current chronopotentiometry was used to characterize the short-term potential stability of GC/POT-CB/K^+^-ISE. Figure 4 shows the typical chronopotentiometric curve of GC/POT-CB/K^+^-ISE. From the same experimental diagram, the short-term stability of the potential can be derived from the ratio ΔE/Δt. The calculated potential drift value of GC/POT-CB/K^+^-ISE is about 10.9 ± 0.5 μV/s, far lower than the value of GC/CB/K^+^-ISM (820.6 ± 46.8 μV/s). According to the equation ΔE/ΔT = I/C (C, low frequency capacitance), the capacitance of the solid contact layer is estimated to be 91.5 μF. The capacitance value is greater than the capacitance value based on graphene SC-ISE reported in literature [44] (75.8 μF) and the capacitance value of SC-ISE based on single-wall carbon nanotubes reported in literature [45] (60 μF). The above results show that the potential stability of the electrode can be significantly improved by using POT-CB as the solid contact layer.

### 2.5. Electrochemical Impedance Spectroscopy

There is a semicircle in the impedance spectrum in the high frequency region. The semicircle is related to the bulk impedance of the ion-selective sensitive membrane, and also to the contact resistance of the solid contact layer and the interface of the ion-selective sensitive membrane [46]. From Figure 5, we can see that the high frequency resistance of GC/POT-CB/K^+^-ISE is about 0.091 MΩ, which is less than the resistance of GC/CB/K^+^-ISE (0.19 MΩ), indicating that the presence of POT-CB can reduce the resistance of electron transfer between the solid contact layer and the interface of the ion-sensitive membrane, and improve the rate of electron transfer at the interface. In the low frequency region, GC/POT-CB/K^+^-ISE has a large semicircle; in contrast, the semicircle of GC/CB/K^+^-ISE in low frequency region is significantly smaller. The main reason is that the existence of POT-CB improves the conductivity between the potassium ion electrode and the ion-sensitive membrane interface and further increases the double-layer capacitance of the electrode.

### 2.6. Water Layer Test

The open circuit potential method was used to measure the water layer and anti-interference performance of the electrode. The test results are shown in Figure 6. A continuous open-circuit potential test on GC/POT-CB/K^+^-ISE in 0.1 M KCl solution was carried out. After 2 h, the electrode was transferred to a 0.1 M NaCl solution for continuous testing for 2 h and then returned to a 0.1 M KCl solution for testing for 8 h. As shown in Figure 6, after transferring to the NaCl solution, the potential of GC/POT-CB/K^+^-ISE drops sharply, but the potential is still very stable. After returning to KCl solution, GC/POT-CB/K^+^-ISE keeps the initial potential value, without drift, and the potential is very stable. This shows that the super-hydrophobicity of the solid contact layer added with POT-CB can effectively avoid the appearance of an interfacial water layer, thus ensuring the excellent potential stability of the electrode

### 2.7. Effects of O_2_, CO_2_ and Light on the Electrode Potential Stability

Figure 7a shows the results of the anti-interference test. The test sequence is O_2_, N_2_, O_2_, N_2_, CO_2_ and N_2_, and each process takes 1 h. From the figure, only a small amount of potential drift can be observed during the conversion of O_2_ and N_2_, and the potential is stable at other times.

Figure 7b shows the light anti-interference test results of GC/POT-CB/K^+^-ISE. It can be seen from the figure that the indoor light and UV light are switched on and off every 30 min. During the switching process, the potential remained stable, showing the strong anti-interference ability of GC/POT-CB/K^+^-ISE.

## 3. Experiment

### 3.1. Experimental Materials

Carbon black (CB), poly (3-octylthiophene-2,5-diyl) (POT), poly (vinyl chloride) (PVC), tetrahydrofuran (THF), bis (2-ethylhexyl) sebacic acid (DOS), valinomycin, tetra—(3,5-bis (trifluoromethyl) phenyl) sodium borate (NaTFPB) and dimethylformamide were purchased from Shanghai McLean Biochemical Technology Co., Ltd.(Shanghai, China). The deionized water with a resistivity of 18.2 M Ωcm was prepared by the Pall Cascada ultrapure water system. Other chemicals used are analytical grade.

### 3.2. Preparation of Solid Contact Layer

Before preparing the electrode, the glassy carbon (GC) electrode (diameter 3 mm) is polished with 1.0, 0.3 and 0.05 μm alumina. Then it is put into deionized water and a deionized water/ethanol mixture for ultrasonic treatment; each process lasts for 5 min. The electrode is finally rinsed with water and dried with nitrogen.

By slightly modifying the procedure reported in literature [47], 10 mg CB was added to 1 mL of poly (3-octylthiophene-2,5-diyl) THF dispersion with a concentration of 10 mg/mL; the obtained suspension was centrifuged (30 min, 7000 rpm), the supernatant was removed and precipitated with ethanol 3 times. The separated solid material (POT-CB) was dispersed in a new 1 mL THF. A drop of 10 μL of POT-CB suspension was placed onto the surface of glassy carbon (GC) electrode and left in the laboratory atmosphere for THF evaporation.

As a contrast, an ion selective electrode using CB as a solid contact layer was made. The CB powder was dispersed in the mixture of dimethylformamide/water in the ratio of 1:1 (*v*/*v*), and the final concentration was 1 mg/mL. Specifically, 10 mg of CB was immersed in 5 mL of dimethylformamide, then 5 mL of water was added and then the dispersion was ultrasonically treated at 60 kHz for 60 min. A drop of 10 mL of CB suspension was placed onto the surface of glassy carbon (GC) electrode and left in the laboratory atmosphere for evaporation.

### 3.3. Preparation of Potassium Ion Selective Electrode

The preparation of K^+^ membrane was based on the research of E Jaworska et al [48], The composition of the potassium ion selective membrane (K^+^-ISM) mixture used in this work (by weight) is about 1.3% tetra (3,5-bis (trifluoromethyl) phenyl) sodium borate (NaTFPB), 2.8% valinomycin, 64.3% DOS and 31.6% PVC. A total of 100 mg of the mixture was dissolved in 1 ml of THF.

A total of 10 μL of K^+^-ISM mixture was dropped onto the dried POT-CB solid contact layer electrode, and the prepared electrode is expressed as GC/POT-CB/K^+^-ISE. At the same time, 10 μL of K^+^-ISM mixture is dropped onto the dried CB solid contact layer electrode, and the prepared electrode is expressed as GC/CB/K^+^-ISE. For comparison, the K^+^-ISM mixture was dropped on the bare GC electrode, and the prepared electrode was expressed as GC/K^+^-ISE. Dry the three electrodes at room temperature for 12 h.

### 3.4. Morphological Characterization

Scanning electron microscope (SEM) imaging of POT-CB and CB solid state transition layers was obtained by a Helios G4 CX focused ion/electron double-beam electron microscope, and the acceleration voltage was 10 kV. The water static contact angle image was measured by the German KRUSS standard DSA25 contact angle measuring instrument.

### 3.5. Electrochemical Measurement

The performance of modified and unmodified electrodes was measured by cyclic voltammetry (CV) in a 5 mM iron/ferrocyanide standard solution, and the equipment model was a CHI760E double potentiostat (Shanghai, China). A three-electrode detection system is adopted, in which a platinum wire electrode is the auxiliary electrode, a potassium ion selective electrode is the working electrode and Ag/AgCl 3 M KCl is the reference electrode. The potential is scanned from −1.0 V to +1.1 V, and the scanning rate is 30 mV/s.

The electrochemical impedance and timing potential of the prepared electrode were measured in a 0.1 M KCl solution. The measurement frequency range of electrochemical impedance spectroscopy is 100 kHz–10 mHz, and the excitation amplitude is 10 mV. The measurement of reverse chronopotential is achieved by applying a constant current (1 nA) in one direction of the ion-selective electrode and keeping it for 60 s, then applying the same time and the same amount of current in reverse and recording the electrode potential–time curve in the whole process.

In the above test process, the prepared potassium ion selective electrode was activated in 10^−3^ M KCl solution for 24 h before use. The adjustment steps were repeated before each measurement, and all electrodes were adjusted and stored separately.

## 4. Detection of Potassium in Real Samples

Using the prepared electrode as the working electrode, we used the standard addition method to determine the K^+^ concentration in river water samples under open circuit voltage conditions. In this work, nine river water samples (taken from Henan University of Technology, Zhengzhou, China) were collected. A total of 10 mL of each sample was taken and added to 100 mL of pure water. After the potential was stabilized, the measured electromotive force was recorded as E_1_. Then, 10 mL of 1 mol/L K^+^ solution was added, and the measured electromotive force was recorded as E_2_. The K^+^ concentration in the river water was calculated according to Formula (2).
(2)$$C_{x}=\frac{C_{s}V_{s}}{V_{x}}{\left(10^{\Delta \left(E_{2}-E_{1}\right)/s}-1\right)}^{-1}$$
where *C*_x_ is the K^+^ concentration of river water, *V*_x_ is the volume of river water, *C*_s_ is the concentration of standard K^+^ solution, *V*_s_ is the volume of standard K^+^ solution and *S* is the response slope of the electrode. In order to compare the accuracy of the detection results of this method, we used atomic absorption spectroscopy (AAS) to measure the concentration of K^+^. The experimental results are shown in Table 2. If the AAS results are taken as the standard, the accuracy of the test results is above 90%, indicating that GC/POT-CB/K^+^ -ISE is feasible for the determination of K^+^ in real samples.

## 5. Conclusions

In this work, a new composite material is proposed as the ion selective electrode of the solid contact layer. CB has relevant electrochemical characteristics and the advantages of low cost and easy preparation of stable dispersion. POT has good hydrophobicity. By using POT as the dispersant of CB and not using surfactant, the hydrophobicity of the solid contact layer can be improved, and the formation of a water layer can be avoided. The nanocomposites obtained from POT and CB can overcome the shortcomings of solid ion selective electrodes. The potential sensor obtained has good performance, with a sensitivity of 57.6 ± 0.8 mV/facade and a detection limit of 10^−6.2^ M for potassium ions. In addition, the improved electrode has strong stability and anti-interference.

## Figures and Tables

**Figure 1 molecules-28-03242-f001:**
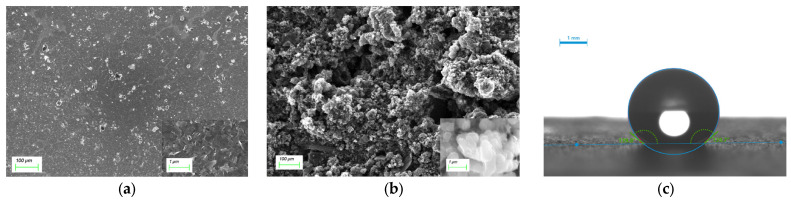
(**a**) Image of CB-ISM solid contact layer, illustration: 1 μm enlarged image; (**b**) image of POT-CB-ISM solid contact layer, illustration: 1 μm enlarged image; (**c**) POT-CB-ISM static contact angle image.

**Figure 2 molecules-28-03242-f002:**
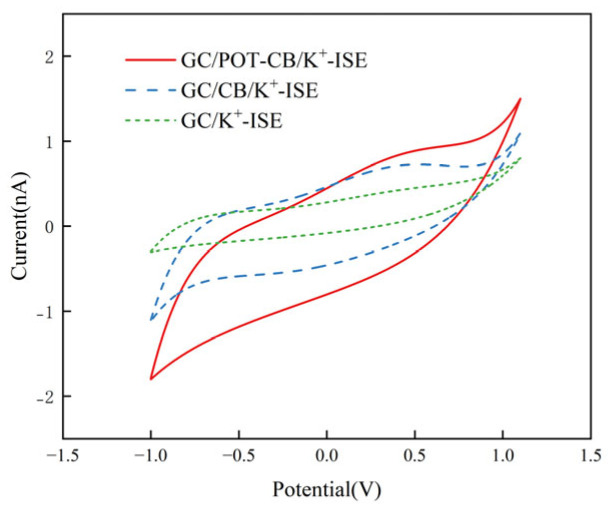
Cyclic Volt–Ampere curve of GC/POT-CB/K^+^-ISE, GC/CB/K^+^-ISE, GC/K^+^-ISE.

**Figure 3 molecules-28-03242-f003:**
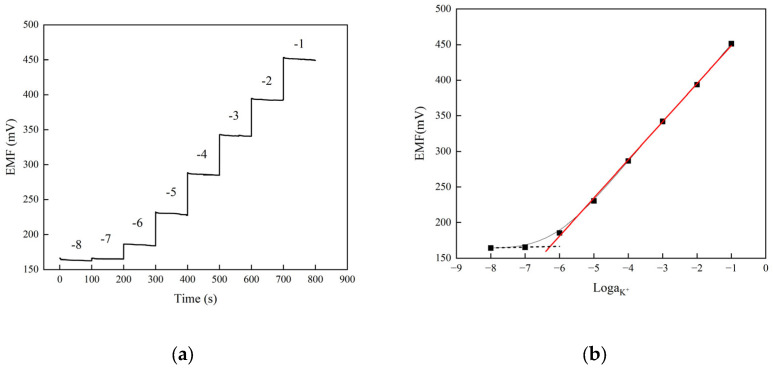
(**a**) Potential response diagram of GC/POT-CB/K^+^-ISE in KCl solution; (**b**) potential calibration curve of GC/POT-CB/K^+^-ISE.The red line is the linear fitting line of the electrode in the range of 10^−6^–10^−1^ M.

**Figure 4 molecules-28-03242-f004:**
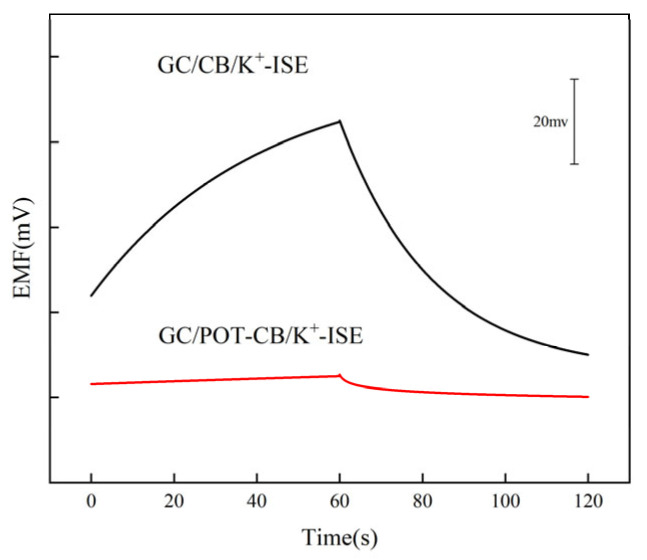
Reverse chronopotentiometric curves of GC/POT-CB/K^+^-ISE and GC/CB/K^+^-ISE.

**Figure 5 molecules-28-03242-f005:**
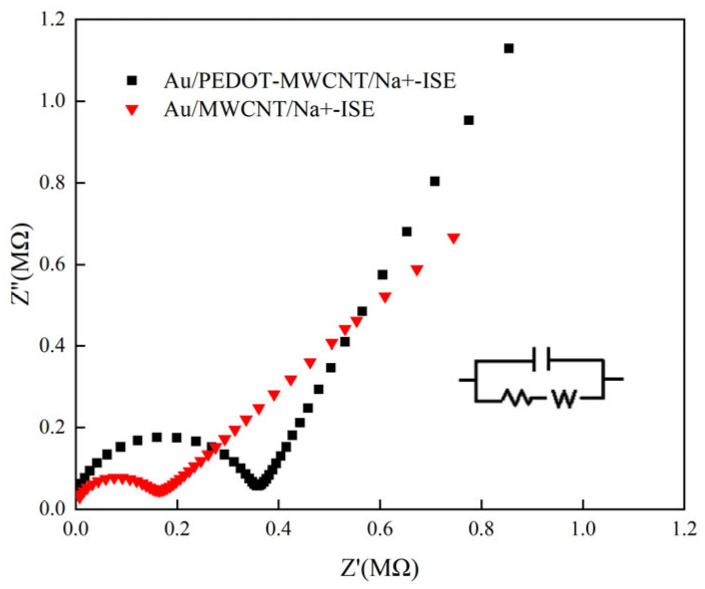
EIS of GC/POT-CB/K^+^-ISE and GC/CB/K^+^-ISE. The illustration is equivalent circuits of GC/POT-CB/K^+^-ISE.

**Figure 6 molecules-28-03242-f006:**
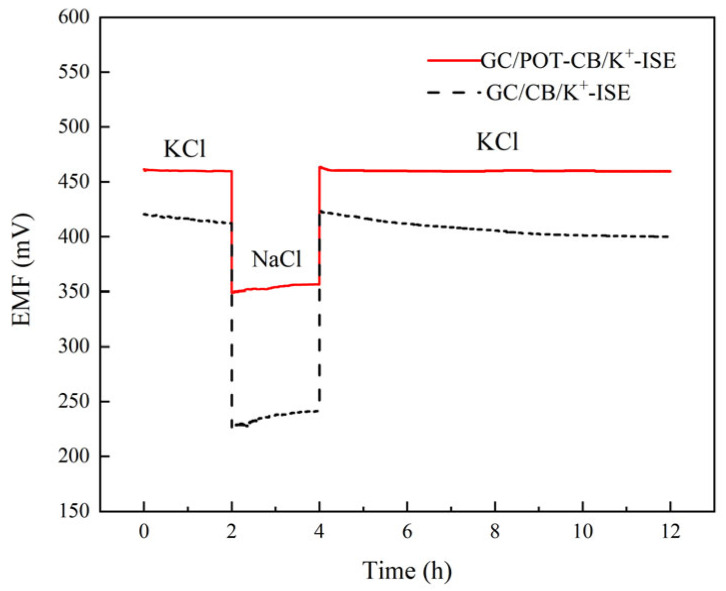
Water layer test diagram.

**Figure 7 molecules-28-03242-f007:**
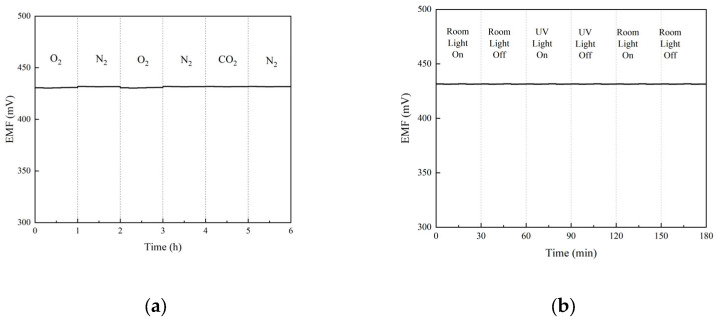
(**a**) The effect of the presence of gases on the operation of sensors; (**b**) the effect of the presence of light on the operation of sensors.

**Table 1 molecules-28-03242-t001:** Summary of Recent Studies of Solid-State Potassium Ion Selective Electrodes.

Electrode	Intermediate Layer/Lonophore	Reference	Sensitivity (mV/decade)	LOD (10^−n^ M)	Linear Range (M)	Ref.
GC	N/A/valinomycin	SCE	49	5	10^−5^–10^−1^	[33]
Pt	Ppy/calixarene	N/A	51	5.7	10^−5.2^–10^−1^	[34]
Pt	PANI/dbdb−18−6	SCE	58	5.8	10^−5^–10^−1^	[35]
Au	PEDOT−PSS/valinomycin	Ag/AgCl 3 M KCl	61.3	3	10^−3^–10^−1.5^	[36]
GC	Graphene/valinomycin	N/A	58.4	6.2	10^−5.8^–10^−1^	[16]
Graphite	Mixture of CB,poly(amidoacid) Cu(I)complex,resin/valinomycin	Ag/AgCl 3.5 M KCl	59	7	10^−6^–10^−1^	[37]
Cu	Graphite-epoxy-hardener/valinomycin	Solid Ag/AgCl	44	4.4	10^−4.3^–10^−1^	[38]
Ag	N/A/PBE	Ag/AgCl 3 M KCl	56.3	4.7	10^−4^–10^−1^	[39]
Pt	Ppy/calixarene	N/A	51	5.7	10^−5.2^–10^−1^	[34]
GC	Hexanethiolate monolayer protected gold cluster/valinomycin	Ag/AgCl 3 M KCl	57.4	6.1	10^−5^–10^−1^	[40]
Pt	Ppy and zeolite/valinomycin	Ag/AgCl 3 M KCl	54.2	5.1	10^−5^–10^−2^	[41]
GC	MoO₂/valinomycin	Ag/AgCl 3 M KCl	55	5.5	10^−5^–10^−3^	[42]
Carbon SPE	PANI/valinomycin	Modified solid Ag/AgCl	60.5	5.8	10^−5^–1	[43]
GC	POT-CB	Ag/AgCl 3 M KCl	57.6	6.2	10^−6^–10^−1^	This Work

**Table 2 molecules-28-03242-t002:** Determination of K^+^ in river water samples (mean ± standard deviation, *n* = 3).

Sample	ISE (m mol/L)	AAS (m mol/L)	Accuracy
1	0.109 ± 0.02	0.119 ± 0.01	91.6%
2	0.111 ± 0.02	0.122 ± 0.01	90.2%
3	0.106 ± 0.01	0.113 ± 0.02	93.8%
4	0.123 ± 0.03	0.134 ± 0.02	91.8%
5	0.116 ± 0.02	0.126 ± 0.02	92.1%
6	0.132 ± 0.03	0.145 ± 0.02	91.0%
7	0.115 ± 0.01	0.127 ± 0.01	90.6%
8	0.125 ± 0.02	0.115 ± 0.02	92.0%
9	0.135 ± 0.03	0.147 ± 0.01	91.8%

## Data Availability

The study did not report any data.

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
