# Peer review of "Conductive Polymer Nanoparticles as Solid Contact in Ion-Selective Electrodes Sensitive to Potassium Ions"

_molecules, 2023, doi:10.3390/molecules28073242_

Round 1

Reviewer 1 Report

In this manuscript, an all-solid potassium ion selective electrode was prepared by poly (3-octylthiophene-2,5-diyl) (POT) and carbon black (CB) nanocomposites serving as ion electron transducer. The potassium ion selective electrode was characterized by a series of electrochemical characterization, which exhibits high sensitivity.

Thus, I recommend this paper to be published after the following questions to be addressed.

[1]    It is best to keep the full image borders in the Figures to improve article’s availability.

[2]    The equivalent circuit image in Figure 5b should be embedded in Figure 5a, and Figures arrangement can be rearranged.

[3]    Whether other material characterizations can be supplemented to further understand the material composition structure?

[4]    Whether the practicability of this new composite material can be explained in the article.

Reviewer 2 Report

The authors have developed a new methodology for detecting potassium by using a new conductive polymer nanoparticle ion selective electrode.

The material is characterized carefully, but the relationship between the structure and electron transfer is not clearly elaborated. So major revision is needed before the acceptance of this manuscript.

Line 111-112

The amounts of materials used for ISE preparation are listed in decimals. Do the authors have some experimental data/graphs that can show the process of optimization of electrode composition? 

“The composition of potassium ion-selective membrane (K+-ISM) mixture used in this work (by weight) is about 1.3% tetra [3,5-bis (trifluoromethyl) phenyl] sodium borate (NaTFPB), 2.8% valinomycin, 64.3% DOS and 31.6% PVC. Dissolve a total of 100mg of the 113

the mixture in 1ml of THF.”

Elaborate why tetra [3,5-bis (trifluoromethyl) phenyl] sodium borate (NaTFPB) was the best choice for the detection of potassium. Why authors didn’t use for example tetrakis(4-chlorophenyl)borate or some other compound of similar structure? 

The effective working area of the prepared electrode should be presented.

The authors need to demonstrate the feasibility of the proposed conductive polymer nanoparticle ion selective electrode for the detection of potassium in real samples.

There are inconsistencies in the list of references, please check the list of references carefully.

Round 2

Reviewer 1 Report

  • The manuscript can be acceped.

Reviewer 2 Report

All the Reviewer's comments are taken into account. I recommend publishing the manuscript in its present form.